# Relationship between the Surface Roughness of Biodegradable Mg-Based Bulk Metallic Glass and the Osteogenetic Ability of MG63 Osteoblast-Like Cells

**DOI:** 10.3390/ma13051188

**Published:** 2020-03-06

**Authors:** Pei-Chun Wong, Sin-Mao Song, Pei-Hua Tsai, Yi-Yuan Nien, Jason Shian-Ching Jang, Cheng-Kung Cheng, Chih-Hwa Chen

**Affiliations:** 1School of Biomedical Engineering, College of Biomedical Engineering, Taipei Medical University, Taipei 110, Taiwan; s0925135546@gmail.com; 2Institute of Materials Science and Engineering, National Central University, Taoyuan 320, Taiwan; bear82112760103@gmail.com (S.-M.S.); peggyphtsai@gmail.com (P.-H.T.); gyes975212@gmail.com (Y.-Y.N.); 3Department of Mechanical Engineering, National Central University, Taoyuan 320, Taiwan; 4School of Biomedical Engineering, Shanghai Jiao Tong University, Shanghai 200240, China; 5Department of Orthopedics, Taipei Medical University - Shuang Ho Hospital, New Taipei 235, Taiwan; 6School of Medicine, College of Medicine, Taipei Medical University, Taipei 110, Taiwan; 7Research Center of Biomedical Device, Taipei Medical University, Taipei 110, Taiwan

**Keywords:** Mg-based bulk metallic glass, biodegradable, surface roughness, osteogenetic

## Abstract

Mg-based bulk metallic glass materials have been investigated for their large potential for application in orthopedic implants due to their biocompatibility, low degradation rate, and osteogenetic ability. As an orthopedic implant, initial cell adhesion has been a critical issue for subsequent osteogenesis and bone formation because the first contact between cells and the implant occurs upon the implants surface. Here, we aimed to create Mg-based bulk metallic glass samples with three different surface roughness attributes in order to understand the degradation behavior of Mg-based bulk metallic glass and the adhesion ability and osteogenetic ability of the contact cells. It was found that the degradation behavior of Mg_66_Zn_29_Ca_5_ bulk metallic glass was not affected by surface roughness. The surface of the Mg_66_Zn_29_Ca_5_ bulk metallic glass samples polished via #800 grade sandpaper was found to offer a well-attached surface and to provide a good cell viability environment for Human MG63 osteoblast-like cell line. In parallel, more calcium and mineral deposition was investigated on extracellular matrix with higher surface roughness that verify the relationship between surface roughness and cell performance.

## 1. Introduction

Mg-based materials have been investigated in recent years for their application in the medical field due to their attractive biocompatibility, biodegradability, and osteogenic abilities. For orthopedic implant applications, Mg-based materials have great potential for interference screws, suture anchors, or bioscrews for bone–bone fixation or bone-tendon/ligament fixation. The release of Mg^2+^ ions during Mg-based material degradation is a mechanism known to enhance osteoconductive, osteoinductive, and osteogenesis ability.

As an orthopedic implant, the surface of the metallic implant is the initial interface that contacts with the bone tissue; thus, the surface roughness of the metallic material is an important consideration, especially when analyzing the interaction between the surface roughness of metallic materials and osteoblasts [1,2,3]. For this reason, the surface roughness of implants has been varied and modified by different surface treatments, such as function coating, in an effort to support the cells initial attachment [4,5]. The materials that are commonly used in orthopedic implants, such as Ti alloys and stainless steel, have been investigated for optimal surface roughness, which has been found to affect the adhesion and subsequent osteogenic function of osteoblasts [6,7,8,9]. The degree of surface roughness has been found to have a limitation window for optimal cell attachment; this window varies with the kinds of cells and materials. For example, the titanium with the surface roughness of 3 um was more suitable than 0.5 um for human osteoblasts differentiation [10]. The titanium with 1.03 μm of surface roughness can offer a better cell response for MC3T3-E1 mouse pre-osteoblast cell line in comparison with the smooth surface samples [11].

Mg-based implant materials are subject to degradation by bodily fluids, and the extent of degradation increases over time. For crystalline Mg-based materials, the spalling phenomenon was first demonstrated on the surface of materials. The following intergranular corrosion and intergranular stress corrosion cracking (IGSCC) have been found to lead to failure of the material [12]. The degradation of amorphous Mg-based materials occurs uniformly and homogeneously; however, the surface roughness varies according to the formation of the products of corrosion and degradation vestiges that occur during degradation process [13]. The surface of biodegradable Mg-based materials was not found to be stable surface because of the surface roughness variation, which is subject to the degradation process and changes all the time [14]. According to the theory, a rough surface would create a more exposed surface for degradation to occur and this surface would therefore increase the degradation rate [14]. The corrosion behavior of Mg alloy has previously been reported by Alvarez et al. [15] and Walter et al. [16]; these reports have focused on the changes to the surface topography, chemical composition distribution, and pitting corrosion after the immersion test. Unfortunately, there are few studies investigating the relationship between degradation behavior and surface roughness of Mg-based material, or the relationship between osteogenesis ability of cells and surface roughness of Mg-based materials. 

In this study, we created Mg_66_Zn_29_Ca_5_ bulk metallic glass materials of three different surface roughness by polishing with sandpaper in order to compare degradation behavior, cell adhesion ability, cell viability, and osteogenesis ability (Figure 1). We hypothesized that the rougher surface roughness of Mg_66_Zn_29_Ca_5_ bulk metallic glass may provide a suitable surface for cell adhesion and improved osteogenetic ability. The results of these tests demonstrate the behavior of biodegradable Mg-based material over a range of different surface roughness that can be further applied to the design and manufacture process of biodegradable Mg-based orthopedic implants.

## 2. Materials and Methods 

### 2.1. Study Design

The objectives of this study were to analyze the optimal surface roughness for cell adhesion and further osteogenetic capability of MG63 osteoblast-like cells (Bioresource Collection and Research Center, Hsinchu, Taiwan) on Mg_66_Zn_29_Ca_5_ bulk metallic glass. Here, we focused on the adhesion cell morphology and the osteogenic sign of MG63 cells on Mg_66_Zn_29_Ca_5_ bulk metallic glass surface with the different surface roughness; fundamental material properties were also analyzed. In this study, three Mg_66_Zn_29_Ca_5_ bulk metallic glass samples of differing surface roughness were processed via surface treatment using sandpaper polishing (#240, #800, and #2000). 

### 2.2. Sample Preparation and Composition Identification

Mg_66_Zn_29_Ca_5_ bulk metallic glass was prepared by the induction melting and injection casting method under an argon atmosphere. Initially, Mg, Zn, and Ca were melted using the induction melting method to form the Mg_66_Zn_29_Ca_5_ ingot. The ingot of Mg_66_Zn_29_Ca_5_ was then re-melted in a quartz tube and injected into a water-cooled Cu mold under an argon atmosphere to form Mg_66_Zn_29_Ca_5_ BMG plate, and then cut into dimensions of 10 mm × 10 mm × 1.5 mm.

### 2.3. Microstructure Analysis

The amorphous state and phase composition of the Mg_66_Zn_29_Ca_5_ BMG samples with different surface roughness were analyzed by X-ray diffraction (XRD) using a D8AXRD X-ray diffractometer (Bruker, Karlsruhe, Germany) with monochromatic Cu Ka radiation.

### 2.4. Surface Roughness Analysis

The Mg_66_Zn_29_Ca_5_ BMG samples with three different surface roughness attributes were analyzed using an Alpha-step profilometer (KLA-Tencor, Milpitas, CA, USA) to determine the relationship between the actual surface roughness and the different sandpaper grades used to polish to Mg_66_Zn_29_Ca_5_ BMG surface.

### 2.5. Degradation Behavior Investigation

The degradation behavior of the Mg_66_Zn_29_Ca_5_ BMG plate samples with different surface roughness parameters was investigated using the immersion method with simulated body fluid (SBF; 8.0 g NaCl, 0.4 g KCl, 0.14 g CaCl_2_, 0.35 g NaHCO_3_, 0.1 g MgCl_2_·6H_2_O, 0.06 g MgSO_4_·7H_2_O, 0.06 g KH_2_PO_4_, and 0.06 g Na_2_HPO_4_·12H_2_O dissolved in 1 L deionized water) (0.3 mL/mm2) at a temperature of 37 °C, according to the standard ASTM G31-72 [17]. After different immersion periods (0, 1, 2, 3, and 4 weeks), the SBF was removed and the samples were rinsed with deionized (DI) water. After the samples were dried, the weight of the samples and the pH value of the SBF were recorded using an electronic balance (Waga Aanalityczna AS 220.R2, Radwag Wagi Elektroniczne, Radom, Poland) and pH meter (PH500, Clean, New Taipei, Taiwan), respectively. The surface roughness of each of the samples was also measured and recorded.

### 2.6. Cell Adhesion Observation and Spreading Calculation

The cell adhesion and spreading behavior of the MG63 osteoblast-like cells were captured by scanning electron microscopy (SEM) (SU3500; Hitachi, Tokyo, Japan). The MG63 osteoblast-like cells were cultured in high-glucose Dulbecco’s modified Eagle medium (DMEM; Gibco^®^, Carlsbad, CA, USA) supplemented with 10% fetal bovine serum (Gibco^®^, Carlsbad, CA, USA). The Mg_66_Zn_29_Ca_5_ BMG plates were placed in a 48-well culture plate, after which 3000 cells/well (2727 cells/cm^2^) MG63 osteoblast-like cells were seeded into the well and subjected to 24 h of incubation in an incubator set at 37 °C in an atmosphere of 5% CO_2_. Prior to observations, fixation, and dehydration processes were used to prepare the samples for SEM. The cell spreading area present on the surface of the Mg_66_Zn_29_Ca_5_ BMG plates was calculated and quantified using Image J software in order to determine the relationship between the different surface roughness attributes of the Mg_66_Zn_29_Ca_5_ BMG samples and cell adhesion ability.

### 2.7. Cell Viability Test (MTT Assay)

Cell viability tests were performed on the samples via the direct contact method. The placement of the materials and the cell culture process was undertaken as per the cell adhesion test. After three days incubation, the Mg_66_Zn_29_Ca_5_ BMG plates were located to a new 48-well culture plate to ensure that the cells that were analyzed all adhered to the surface. Ten microliters of MTT (3-(4,5-dimethylthiazol-2-yl)-2,5-diphenyltetrazolium bromide) solution (Invitrogen, Carlsbad, CA, USA) was then added carefully to each well, and incubation was done for a further 3 h. One-hundred microliters of dimethylsulfoxide was then placed into each well, and the optical density value was detected using an enzyme-linked immune-sorbent assay reader at a wavelength of 560 nm (Multiskan FC; Thermo, Waltham, MA, USA).

### 2.8. Extracellular-matrix Calcium Deposition Detection

MG63 osteoblast-like cells were used to facilitate extracellular-matrix calcium deposition by the ions released from the Mg_66_Zn_29_Ca_5_ BMG plates with different surface roughness. The Mg_66_Zn_29_Ca_5_ BMG plates with different surface roughness were treated with the cell culture process through the same method as for the cell adhesion test. After incubation for 7 days, the Mg_66_Zn_29_Ca_5_ BMG plates were transferred to another 48-well culture plate. Cells were fixed using 4% paraformaldehyde, and Alizarin Red S (ARS), a staining dye, was used to stain the calcium deposits generated by the MG63 cells. After 40 min, the ARS dye was removed and solubilized with 100 μL of DMSO for 30 min, and the optical density was measured at 590 nm with an enzyme-linked immune-sorbent assay reader (Spectra Max 190, Molecular Device, San Jose, CA, USA).

### 2.9. Migration Test

For the migration test, in vitro scratch assay was used to investigate the migration ability of the MG63 osteoblast-like cells. The Mg_66_Zn_29_Ca_5_ BMG plates with different surface roughness were first placed in 48-well culture plate, after which 5000 cells from the cell suspension were suspended in a 48-well culture plate and incubated for 6 h at 37 °C in an atmosphere of 5% CO_2_. After 2 h, cell monolayers formed on the surface of the material. A 1000 μL pipet tip was used to scratch a straight line along the monolayer. The debris was then removed, and the cells were washed gently using DMEM, after which additional DMEM was added to cover the cells for incubation. After 4 h of incubation, the DMEM was removed, and 4% paraformaldehyde was added to facilitate the fixation process and to execute the dehydration procedure for the SEM observation. Images were captured using SEM (SU3500; Hitachi, Tokyo, Japan), and the distance traveled during the desired time frame was measured by Image J to determine the migration ability (Figure 2).

### 2.10. Statistical Analysis

All results in this study were presented as mean ± standard deviation. All statistical analyses were performed by SPSS (version 20.0, IBM, Armonk, NY, USA). One-way ANOVA analyses of variance followed by post-hoc Tukey tests and independent sample *t*-test were used to analyze the data collected during this study. The statistical significance was set at a p value of <0.05.

## 3. Results

### 3.1. Degradation Behavior

The degradation behavior of the Mg_66_Zn_29_Ca_5_ bulk metallic glass samples with three different surface roughness parameters is shown in Figure 3; it includes the change in the pH value of the simulated body fluid, the weight change, and the change in the surface roughness. The samples with three different surface roughness demonstrated similar behavior with respect to pH change of the SBF and weight change after 4 weeks of immersion (Figure 3a,b). According to the results, there was no relationship between the change in surface roughness of the samples and the corresponding degradation behavior. The surface roughness of all samples was observed to change in the SBF during immersion. The original surface roughness values of samples, which were subjected to grinding by #240, #800, and #2000 sandpapers, were 0.24, 0.22, and 0.07 μm, respectively. After 4 weeks of immersion, the surface roughness was found to increase to 1.01, 0.74, and 0.74 μm, respectively (Figure 3c).

### 3.2. Microstructure Characterization

The XRD patterns of the Mg_66_Zn_29_Ca_5_ bulk metallic glass samples with three different surface roughness parameters prior to and after four weeks of immersion are shown in Figure 4. Prior to immersion, all sample groups showed a typical broad diffraction peak around 30°–50° of glassy matrix with a Mg_0.97_Zn_0.03_ phase crystalline peak (Figure 4a). However, after degradation by the immersion test, the amorphous matrix decreased, and the crystalline peak then appeared because of the generation of corrosion residues in the samples (Figure 4b).

### 3.3. Observation of Surface Morphologies 

The surface morphologies of the Mg_66_Zn_29_Ca_5_ bulk metallic glass with three different surface roughness parameters prior to and after 4 weeks of immersion are shown in Figure 5. Prior to degradation from immersion, the surfaces were flat, and the grinding trace of the sandpaper were clearly visible. After 4 weeks of immersion, the surfaces of the samples changed, and the roughness increased for all groups. 

### 3.4. Cell Adhesion and Spreading

The cell adhesion morphology on the Mg_66_Zn_29_Ca_5_ bulk metallic glass samples with different surface roughness were captured by SEM after fixation and a dehydration process (Figure 6). The three different surface roughness factors were all found to provide a suitable surface for MG63 cell adhesion with a distinct spindle shape. The well-expanded cytoskeleton of the MG63 cells on the three different surfaces, the formation of the pseudopodia, and the pronounced spreading on the different surface samples demonstrated that the cells were tightly attached (indicated by arrow) (Figure 6a–c). The MG63 cells not only formed two-dimensional structures but also three-dimensional structures, with the many cell–cell interactions able to be clearly visualized (Figure 6d–f). However, different percentages of cell coverage were observed on the different surfaces. The greatest percentage of cell coverage area was observed for the samples from the group ground with the #800-grade sandpaper (Figure 6g), at 80.96%. 

### 3.5. Cell Viability

The cell viability of the MG63 cells (normalized against the control group) cultured on the Mg_66_Zn_29_Ca_5_ bulk metallic glass samples with different surface roughnesses is shown in Figure 7. The surface of the samples polished with the #800-grade sandpaper was observed to have the greatest cell viability when compared with the samples polished with the sandpaper grades. The cell viabilities of the Mg_66_Zn_29_Ca_5_ bulk metallic glass samples polished with the #240, #800, and #2000 grade sandpapers were 76.9 ± 0.2, 91.4 ± 0.3, and 68.8 ± 0.2, respectively. According to ISO 10993-5 [18], the cell viability of the #800 group is classified as first-level cytotoxicity (slight-cell survival rate higher than 80%), and that for the #240 and #2000 groups is classified as second-level (mild-cell survival rate higher than 50%).

### 3.6. Extracellular-matrix Calcium Deposition

The extracellular-matrix calcification of the MG63 cells was detected using Alizarin Red S staining. The quantitative analysis of the ARS dye that solubilized with DMSO after normalization is shown in Figure 8. The calcium deposition rates observed on the samples with different surface roughness was 97.26 ± 2.12% for the #240 group, 84.81 ± 0.99% for the #800 group, and 78.33 ± 3.04% for the #2000 group (Figure 8a). On the other hand, the Mg_66_Zn_29_Ca_5_ bulk metallic glass samples without MG63 cells have no extracellular-matrix calcification on the surface (Figure 8b).

### 3.7. Migration Capacity

To shorten the period required for a bone to heal and to enhance the cell migration velocity for cells to attach to the area surrounding the implanted material, materials with three different surface roughness factors were tested. After gap generation, MG63 cells had 4 h to migrate during incubation. The gap can be seen clearly in Figure 9a as commencing after 4 h of incubation. The different surface roughness attributes of the samples exhibited different gap distances even though each was created using the same pipet tip size. The distance of the gap, which was created by the pipet tip, was reduced through the migration of MG63 cells. The reduced distance was measured using Image J software (results are shown in Figure 9b). It was found that the reduction in the gap distance of the #800 group (255 ± 50 μm) and the #2000 group (201 ± 19 μm) were significantly higher than that observed for the #240 group (66 ± 13 μm). The high magnification of cell migration after 4 h of incubation is shown in Figure 9c.

## 4. Discussion

Mg_66_Zn_29_Ca_5_ bulk metallic glass is an amorphous structure material that can be considered as a single-crystal material due to its homogeneous and uniform arrangement. Thus, the Mg_66_Zn_29_Ca_5_ bulk metallic glass material remains as a single crystal structure after being polished by different grades of sandpaper. The retention of the crystal structures is responsible for the lack of difference in the degradation behavior observed for the Mg_66_Zn_29_Ca_5_ bulk metallic glass samples of varying surface roughness, as seen in Figure 3a,b. However, the surface roughness varied greatly after 4 weeks of degradation (Figure 3c). The variation in the roughness of the samples surface was not observed to influence the recorded change in weight and pH value, which remained stable. As such, the surface roughness change caused by degradation of the samples occurred uniformly and homogeneously. Compared with Mg crystalline alloy, AZ91 magnesium alloy, the pitting corrosion increased the progress of degradation in which the stress corrosion crack contributed to material failure [16]. Moreover, the microstructure of the metallic glass gradually crystallized as degradation time increased [13]. From our results, Mg_66_Zn_29_Ca_5_ bulk metallic glass was observed to exhibit decreased amorphous phase after 4 weeks of immersion. It should be noted that the surface of the samples polished with #240 grade sandpaper was observed to have the greatest crystalline signal in the XRD pattern (Figure 4b); presumably, the corrosion products more easily formed and cohered on the rough surface. Nguyen et al. previously reported that the corrosion rate observed for rough surface samples of as-cast Mg was significantly higher than that for smooth samples [19]. This higher corrosion rate may lead to more corrosion products being generated on the rough surface.

The surface of implant materials is the first site in contact with the surrounding cells and tissue. In this study, for Mg-based BMG of the same composition, the surface roughness was found to be the only factor determining the biocompatibility and cell adhesion of MG63 cells to Mg_66_Zn_29_Ca_5_ bulk metallic glass. Recently, orthopedic implants have been used as functional coating or to increase surface roughness to enhance the resulting biocompatibility, biological fixation, and osteoconductive properties [20,21]. However, the surface of Mg materials changes over time because of their biodegradable properties. Thus, using physical methods (e.g., grinding) to treat the original metallic surface may be a direct way of improving the biological effectiveness for cells without requiring any coating. According to the results in this study, a surface roughness of around 0.22 μm (#800 group) was shown to result in the greatest cell coverage area after 24 h of incubation (Figure 6). After 3 days of incubation, the #800 group also showed the highest cell viability (Figure 7). A surface roughness of 0.22 μm therefore seems to provide a good platform for initial cell adhesion and survival. The entrapment of fibrin protein has been encouraged by the rough surface, increased the adhesion of osteogenic cells and enhanced the mechanical stability of implants in human bone, our work [22,23,24,25,26,27,28]. To facilitate osteointegration, the dense bone tissue has to entirely cover the surface of the material. The materials should not only have good cell proliferation but also be effective in recruiting cells from the surrounding tissue to achieve this ideal situation. In the results obtained from our study, the samples polished using the #800-grade sandpaper demonstrated the best migration ability for the MG63 cells. However, the most effective calcium deposition was observed for the samples polished with the #240 grade sandpaper; unfortunately, #240 group showed the worst cell migration ability. 

When considering the surface roughness of material designed for orthopedic implants, the surface should be varied according to the different stages of osteointegration. For example, materials with a surface roughness of #800 are best for initial cell adhesion, cell migration, and cell proliferation; an increased surface roughness will support improvements in the subsequent calcium deposition that is required for bone cells. The surface roughness variation of biodegradable magnesium material during degradation may have a strong advantage over other metal materials, which cannot be degraded when implanted in the human body.

## 5. Conclusions

In this study, Mg_66_Zn_29_Ca_5_ bulk metallic glass samples treated to produce samples with three surface roughness parameters were investigated using material properties test and in vitro study. We found that different surface roughness factors can enhance different functions for MG63 cells. However, there was no significant difference between the surface roughness of the three samples in terms of degradation behavior. For orthopedic implant applications, harnessing different surface roughness produced by degrading materials in order to regulate different functions of cells should be considered. The variation of the surface roughness of implant material can be achieved via natural or engineering processes. For the design of surface treatment, the surface roughness which polishes with #800-grade sandpaper would be recommended to provide the surface for cell attachment. The much rougher surface would be generated naturally during degradation to provide the surface for cells to enhances the effects of calcium deposition.

## Figures and Tables

**Figure 1 materials-13-01188-f001:**
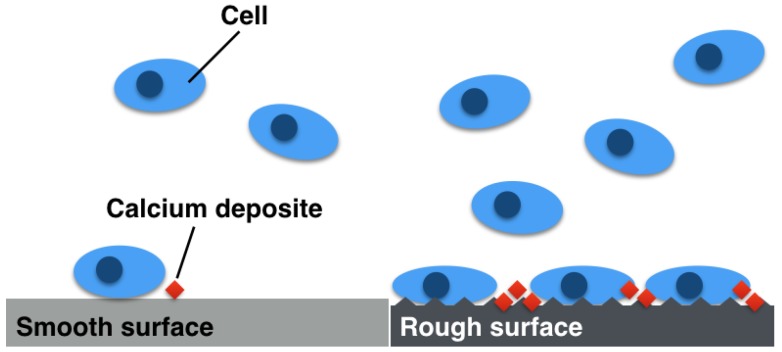
The relationship between Mg_66_Zn_29_Ca_5_ bulk metallic glass with varying surface roughness and MG63 cell adhesion, and the following calcium extracellular matrix deposition.

**Figure 2 materials-13-01188-f002:**
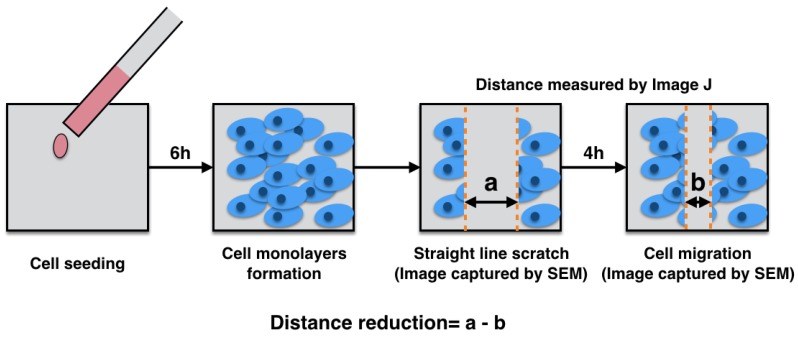
The migration test procedure.

**Figure 3 materials-13-01188-f003:**
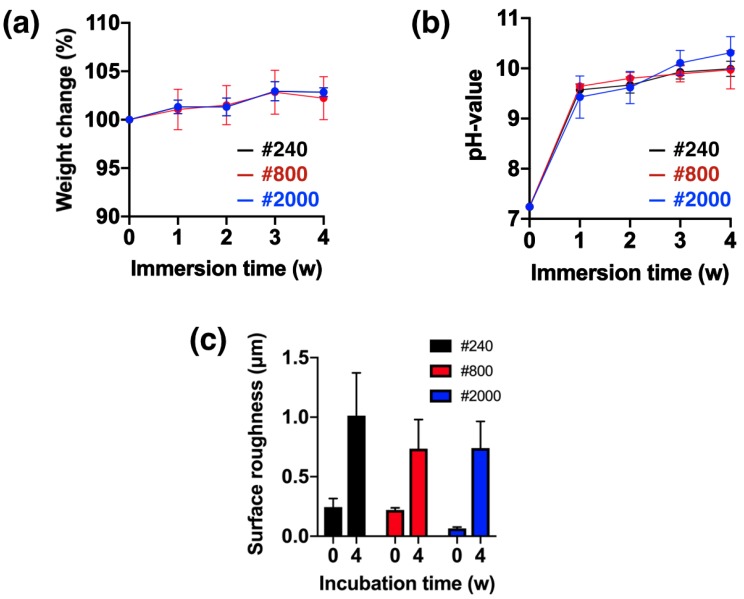
Behavior of Mg_66_Zn_29_Ca_5_ bulk metallic glass samples with three different surface roughness attributes. (**a**) pH change of simulated body fluid, (**b**) weight loss of sample, and (**c**) surface roughness change.

**Figure 4 materials-13-01188-f004:**
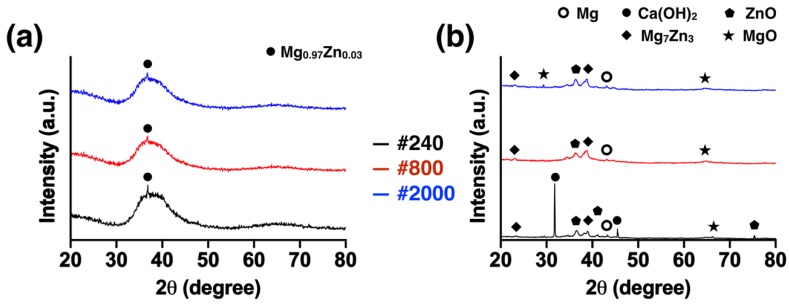
Pattern of Mg_66_Zn_29_Ca_5_ bulk metallic glass samples with three different surface roughness parameters (**a**) prior to the immersion test and (**b**) after 4 weeks of immersion.

**Figure 5 materials-13-01188-f005:**
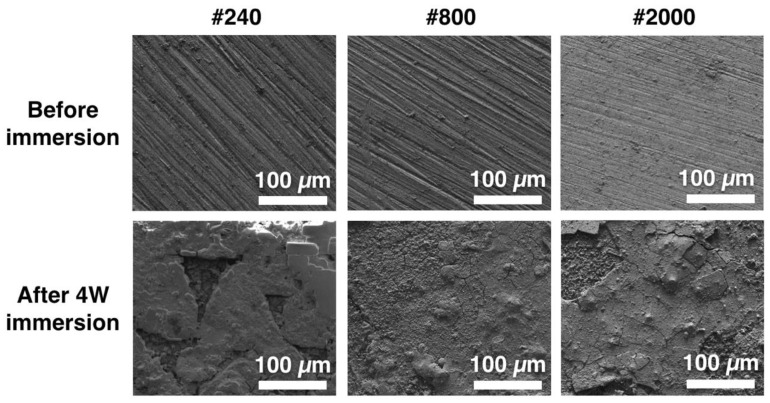
Morphology observations of Mg_66_Zn_29_Ca_5_ bulk metallic glass samples with three different surface roughness parameters before and after the four-week immersion test.

**Figure 6 materials-13-01188-f006:**
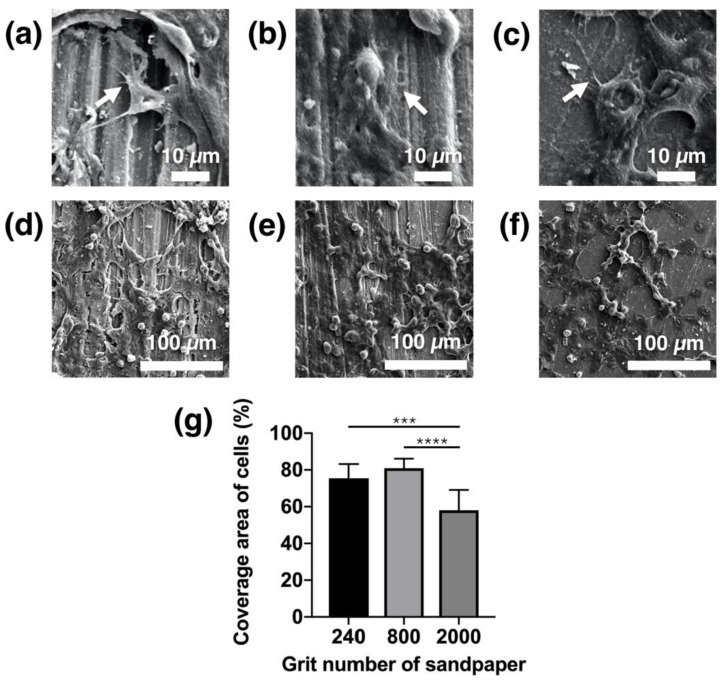
Cell adhesion observed on the Mg_66_Zn_29_Ca_5_ bulk metallic glass samples with different surface roughness. SEM images of the cells, which attached on the surfaces polished using (**a**,**d**) #240, (**b**,**e**) #800, and (**c**,**f**) #2000 grades of sandpaper; (**g**) quantified adhesion of MG63 cells. The cytoskeleton of the MG63 cells was well-expanded, and spreading and formation of pseudopodia on the surface of the Mg_66_Zn_29_Ca_5_ bulk metallic glass was apparent (indicated by the arrow). (* *p* < 0.05, ** *p* < 0.01, *** *p* < 0.005, and **** *p* < 0.001).

**Figure 7 materials-13-01188-f007:**
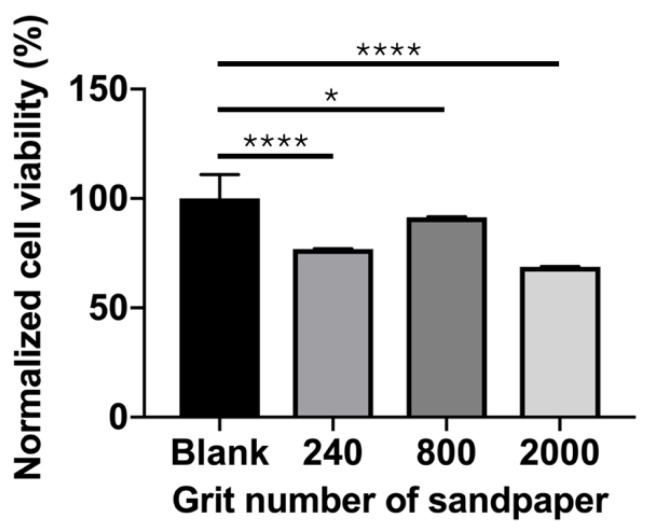
Viability of the MG63 cells, which were cultured on the different Mg_66_Zn_29_Ca_5_ bulk metallic glass surface samples for three days. (* *p* < 0.05, ** *p* < 0.01, *** *p* < 0.005, and **** *p* < 0.001).

**Figure 8 materials-13-01188-f008:**
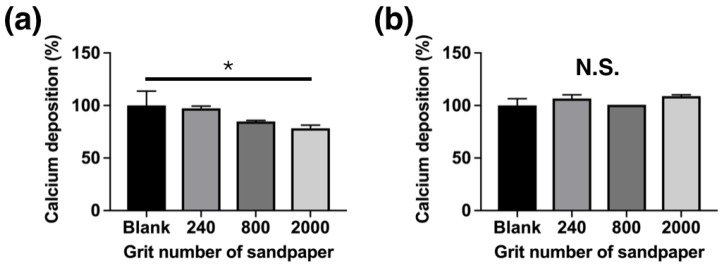
Quantitative results for (**a**) extracellular-matrix calcium and mineral deposition by MG63 cells cultured on the surface of the Mg_66_Zn_29_Ca_5_ bulk metallic glass samples. (**b**) without MG63 cells cultured on the surface of the Mg_66_Zn_29_Ca_5_ bulk metallic glass sample. Alizarin red S staining method was used in this test. (N = 5 per group; * *p* < 0.05, ** *p* < 0.01, *** *p* < 0.005, and **** *p* < 0.001).

**Figure 9 materials-13-01188-f009:**
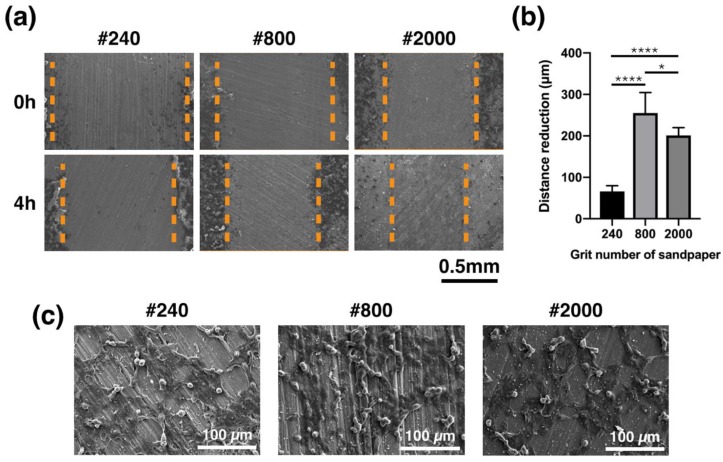
Capacity of MG63 cells cultured on Mg_66_Zn_29_Ca_5_ bulk metallic glass samples with three different surface roughness characteristics. (**a**) SEM image of migrated cells. (**b**) Distance reduction of the gap due to migration of MG63 cells. (**c**) High magnification image of cell migration morphology after 4 h of incubation. (* *p* < 0.05, ** *p* < 0.01, *** *p* < 0.001, and **** *p* < 0.001).

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
