# Peer review of "Relationship between the Surface Roughness of Biodegradable Mg-Based Bulk Metallic Glass and the Osteogenetic Ability of MG63 Osteoblast-Like Cells"

_materials, 2020, doi:10.3390/ma13051188_

Round 1

Reviewer 1 Report

The paper is well-written and shows new results. However, these are some comments and questions for the authors' consideration:

Why only 3 roughness levels were chosen for this study? Would not investigating more levels (especially between 800 to 2000) give more confidence to the drawn conclusions? The pH of the human body is around 7.4, why did the pH was measured and kept increasing during the test? The conclusion section is very short, it needs to be expanded.

Author Response

#1.   Why only 3 roughness levels were chosen for this study? Would not investigating more levels (especially between 800 to 2000) give more confidence to the drawn conclusions?

Response: Thank you for your comments. In this study, the relationship between the range of surface roughness of materials and the cell attachment, and further biological effect is what we like to understand. Three roughness level can help us to understand the range of surface roughness which can be a potential condition for surface treatment.

#2.   The pH of the human body is around 7.4, why did the pH was measured and kept increasing during the test?

Response: Thank you for the comment. The pH value will increased because of the Mg ions and hydrogen release during degradation.

#3.   The conclusion section is very short, it needs to be expanded.

Response: Thank you for your suggestion, the sentences have been added in the conclusion part.

Reviewer 2 Report

Introduction

I refer to Sentence: The materials that are commonly used in orthopedic implants, such as Ti alloys and stainless steel, have been investigated for optimal surface roughness, which has been found to affect the adhesion and subsequent osteogenic function of osteoblasts [6-9]. - It is clear. But Ti alloys are neutral implants, and Mg alloys are a potential material for resorbable implants. It should be noted that the assumed roughness will change very much as degradation progresses. Therefore, the roughness value that the author assumes at the initial stage of degradation may be beneficial, while in our progress the degradation will not matter. I refer to Sentence: The degree of surface roughness has been found to have a limitation window for optimal cell attachment; this window varies with the kinds of cells and materials. - Please provide examples of these optimal ranges depending on the material and tissues. In this way, it is easier to expect the purpose of this work. What is the roughness value the author of the paper is seeking ?? I refer to Sentence: We hypothesized that the optimal surface roughness of Mg66Zn29Ca5 bulk metallic glass may provide a suitable surface for cell adhesion and improved osteogenetic ability. - this is not a hypothesis.

Results

Table 1. Chemical composition of Mg66Zn29Ca5 bulk metallic glass with three different surface 180 roughness attributes - why do you need this result ?? what is the purpose of this study??

4.3 Microstructure characterization

FIG.3a - THIS IS NOT AMORPHOUS MATERIAL. In figure 3a, there is a VISIBLE PEAK FROM PHASE MG0.97Zn0.03. If the authors claim that the studied material is amorphous, please support it with other tests, e.g. TEM. Why fig.3.b does not describe where the peaks come from?? - , no phase analysis

4.6 Cell viability

I refer to sentence - According to ISO 10993-5 [16], the 246 cell viability of the #800 group is classified as first-level cytotoxicity (slight), and that for the #240 and 247 #2000 groups is classified as second-level (mild). –please explain what means

Conclusions

Please summarize precisely what effect roughness has on the degradation of the alloys tested. What is the most favorable roughness

Author Response

#1. I refer to Sentence: The materials that are commonly used in orthopedic implants, such as Ti alloys and stainless steel, have been investigated for optimal surface roughness, which has been found to affect the adhesion and subsequent osteogenic function of osteoblasts [6-9]. - It is clear. But Ti alloys are neutral implants, and Mg alloys are a potential material for resorbable implants. It should be noted that the assumed roughness will change very much as degradation progresses. Therefore, the roughness value that the author assumes at the initial stage of degradation may be beneficial, while in our progress the degradation will not matter.

Response: Thank you for your useful and incisive comment. For the biodegradable/ neutral biomaterials, the initial surface roughness of material was the key factor to determine that the initial attachment level of cells. For biodegradable Mg-based materials, however, the surface roughness would be changed during degradation and the cell response would also be changed. This is what we want to understand and investigate.

#2. I refer to Sentence: The degree of surface roughness has been found to have a limitation window for optimal cell attachment; this window varies with the kinds of cells and materials. - Please provide examples of these optimal ranges depending on the material and tissues. In this way, it is easier to expect the purpose of this work. What is the roughness value the author of the paper is seeking ??

Response: Thank you for your suggestion. A few sentences have been added to the updated manuscript.

#3. I refer to Sentence: We hypothesized that the optimal surface roughness of Mg66Zn29Ca5 bulk metallic glass may provide a suitable surface for cell adhesion and improved osteogenetic ability. - this is not a hypothesis.

Response: Thank you for the comment. The sentence has been changed in row 72 in the updated manuscript.

#4. Table 1. Chemical composition of Mg66Zn29Ca5 bulk metallic glass with three different surface 180 roughness attributes - why do you need this result ?? what is the purpose of this study??

Response: Chemical composition check is the primary test to ensure the composition of materials what we prepared are same as our design. The table 1. has been remove in this manuscript.

#5. FIG.3a - THIS IS NOT AMORPHOUS MATERIAL. In figure 3a, there is a VISIBLE PEAK FROM PHASE MG0.97Zn0.03. If the authors claim that the studied material is amorphous, please support it with other tests, e.g. TEM. Why fig.3.b does not describe where the peaks come from?? - , no phase analysis

Response: Thanks for your comments. About figure 3a, the XRD pattern shows the typical broaden diffraction peak around 30°–50° of glassy matrix and accompany with some crystalline peaks which corresponding to the crystalline phases embedded in the amorphous matrix. Our previous publication has shown a similar result about the XRD pattern (Wong et al. Degradation behavior and mechanical strength of Mg-Zn-Ca bulk metallic glass composites with Ti particles as biodegradable materials. Journal of alloys and compounds 2017, 699, 914-920.). The phase analysis are shown in an updated figure 3b.

#6. I refer to sentence - According to ISO 10993-5 [16], the 246 cell viability of the #800 group is classified as first-level cytotoxicity (slight), and that for the #240 and 247 #2000 groups is classified as second-level (mild). –please explain what means

Response: According to ISO 10993-5, first-level (slight) cytotoxicity means the cell survival rate higher than 80% and second-level (mild) cytotoxicity means the cell survival rate higher than 50%.

#7. Please summarize precisely what effect roughness has on the degradation of the alloys tested. What is the most favorable roughness

Response: Thank you for the suggestion. We have added a few sentences in the conclusion to illustrated the degradation behavior of samples and the most favorable roughness.

Reviewer 3 Report

The scientific paper deals with the relatively simple idea of improving  adherence of osteoblast-like cells by modifying the Mg66Zn29Ca5 bulk
metallic glass surface by grinding. The idea is simple, but the practical conclusion drawn from this article is significant and worth for publication. According to the results, surface roughness of around 0.22 μm shows the highest cell viability and adhesion .

Author Response

#1. The scientific paper deals with the relatively simple idea of improving adherence of osteoblast-like cells by modifying the Mg66Zn29Ca5 bulkmetallic glass surface by grinding. The idea is simple, but the practical conclusion drawn from this article is significant and worth for publication. According to the results, surface roughness of around 0.22 μm shows the highest cell viability and adhesion.

Response: Thank you for your kindly comment.

Reviewer 4 Report

Dear Authors,

The work you accomplished is discretely presented, the aim behind the work is clear although does not add any significant step forward in the biomaterials field. You discussed the influence of 3 different roughnesses of the same material on several parameters, although the discussion seems to me sometimes . The fact that surface roughness. Beside the scientific content, then there are few missing parts and mistakes, here some I'm going to provide you some comments to improve the quality of the manuscript:

1) I would change in the abstract the last sentence:

"The findings in this study indicate that varying the surface roughness of implants influences cell function, for which the single surface does not enhance all of the functions for cells." There is the repetition of cell function and the sentence is to me not very clear, I would therefore reformulate it in a more clear way. 

2) In the materials part the sentence: "Mg66Zn29Ca5 bulk metallic glass samples of differing surface roughness were processed via surface treatment using sandpaper polishing (#240, #800, and #2000) row 87" is then repeated few raws ahead (raw 95), i would therefore omit one of the two. 

3)Row 176 i would move it to materials and methods

4) Section numbering is wrong, skips from 3.1 to 4.2, please correct. 

5) Row 175 mentions compressive strength although is not present in the whole manuscript. Also, unit measure in the table are missing? What is the %? Atomic? molar? 

6) Row 268-69 Row materials with three different surface roughness factors were tested to determine the impact of the different surfaces on the migration ability of MG63 cells. It's here clear you produced 3 materials with variable roughness to be tested in different aspects, should be omitted. 

I would add some explanation about profilometry technique, how did you obtain those numbers, maybe as supplementary material.

7)Sentences as "using surface roughness to treat the original metallic surface" have to be rewritten, surface roughness is the result of the treatment, it's not something you use to treat the material...there are several sentences of this kind and they have to be modified since they impair the overall quality of the article.

8) Fig. 5. I would change the order to Figures so as to ease the comparison between them, now they are paired in the caption (a, d) (b, e) etc.

Best regards

Author Response

#1. I would change in the abstract the last sentence: "The findings in this study indicate that varying the surface roughness of implants influences cell function, for which the single surface does not enhance all of the functions for cells." There is the repetition of cell function and the sentence is to me not very clear, I would therefore reformulate it in a more clear way.

Response: Thank you for your useful suggestion, the sentence has been changed in the abstract.

#2. In the materials part the sentence: "Mg66Zn29Ca5 bulk metallic glass samples of differing surface roughness were processed via surface treatment using sandpaper polishing (#240, #800, and #2000) row 87" is then repeated few raws ahead (raw 95), i would therefore omit one of the two.

Response: Thank you for the correction. The sentences in section 2.2 have been removed.

#3. Row 176 i would move it to materials and methods

Response: The section “3.1 Chemical composition and compressive strength” has been remove.

#4. Section numbering is wrong, skips from 3.1 to 4.2, please correct.

Response: Thank you for the correction. The section number has been rearranging.

#5. Row 175 mentions compressive strength although is not present in the whole manuscript. Also, unit measure in the table are missing? What is the %? Atomic? molar?

Response: The section “3.1 Chemical composition and compressive strength” has been remove and the section number has been rearranging. Moreover, the percentage means atomic percentage.

#6. Row 268-69 Row materials with three different surface roughness factors were tested to determine the impact of the different surfaces on the migration ability of MG63 cells. It's here clear you produced 3 materials with variable roughness to be tested in different aspects, should be omitted.

Response: The sentence has been omitted in the updated section 3.7.

#7. Sentences as "using surface roughness to treat the original metallic surface" have to be rewritten, surface roughness is the result of the treatment, it's not something you use to treat the material...there are several sentences of this kind and they have to be modified since they impair the overall quality of the article.

Response: Thank you for your useful suggestion. The sentence has been changed in the updated manuscript. (row 296)

#8. Fig. 5. I would change the order to Figures so as to ease the comparison between them, now they are paired in the caption (a, d) (b, e) etc.

Response: Thank you for the suggestion. Figure 5 has been rearranging in this updated manuscript.

Round 2

Reviewer 2 Report

I accept responses.

Author Response

Thank you.

Reviewer 4 Report

Dear Authors,

Thank you for the revised versions, although there is some improvement, still some revision is required, especially for what concerns the quality of written English, the expression of concepts and ideas. Often the sentences are not clear because of this reason, sometimes badly formulated, sometimes incomplete. Moreover, often some terms are repeated with no added value, but on the contrary, rendering tedious the whole reading experience. I also decided to remove in some parts words as "systematic" cause I don't believe that the work you performed could be defined as systematic, since only 3 roughness  values were discussed.

In caption 5 and 6 there is reference to N, p, * , ** etc...but I couldn't understand what do they refer to, I tried to look for it in the text but couldn't find it. Please in case you have omitted it, specify and explain something in the text.

Best regards

Author Response

#1.   Thank you for the revised versions, although there is some improvement, still some revision is required, especially for what concerns the quality of written English, the expression of concepts and ideas. Often the sentences are not clear because of this reason, sometimes badly formulated, sometimes incomplete. Moreover, often some terms are repeated with no added value, but on the contrary, rendering tedious the whole reading experience.

Response: Thank you for your suggestion. We have edited the manuscript.

#2.   I also decided to remove in some parts words as "systematic" cause I don't believe that the work you performed could be defined as systematic, since only 3 roughness  values were discussed.

Response: Thank you for your comment. I have removed the term “systematic” in this manuscript.

#3.   In caption 5 and 6 there is reference to N, p, * , ** etc...but I couldn't understand what do they refer to, I tried to look for it in the text but couldn't find it. Please in case you have omitted it, specify and explain something in the text.

Response: Thank you for your comment. In caption 5 and 6, n means the sample size. p means p-value, the p-value is the probability of obtaining results as extreme as the observed results of a statistical hypothesis test. I have removed “n”, but I have to keep “p” in the manuscript to illustrate the significant difference.